# Cooperation of Various Cytoskeletal Components Orchestrates Intercellular Spread of Mitochondria between B-Lymphoma Cells through Tunnelling Nanotubes

**DOI:** 10.3390/cells13070607

**Published:** 2024-03-30

**Authors:** Henriett Halász, Viktória Tárnai, János Matkó, Miklós Nyitrai, Edina Szabó-Meleg

**Affiliations:** 1Department of Biophysics, Medical School, University of Pécs, H-7624 Pécs, Hungary; 2Department of Immunology, Faculty of Science, Eötvös Loránd University, H-1117 Budapest, Hungary

**Keywords:** actin, microtubule, mitochondria, transport, membrane nanotube, motor proteins

## Abstract

Membrane nanotubes (NTs) are dynamic communication channels connecting spatially separated cells even over long distances and promoting the transport of different cellular cargos. NTs are also involved in the intercellular spread of different pathogens and the deterioration of some neurological disorders. Transport processes via NTs may be controlled by cytoskeletal elements. NTs are frequently observed membrane projections in numerous mammalian cell lines, including various immune cells, but their functional significance in the ‘antibody factory’ B cells is poorly elucidated. Here, we report that as active channels, NTs of B-lymphoma cells can mediate bidirectional mitochondrial transport, promoted by the cooperation of two different cytoskeletal motor proteins, kinesin along microtubules and myosin VI along actin, and bidirectional transport processes are also supported by the heterogeneous arrangement of the main cytoskeletal filament systems of the NTs. We revealed that despite NTs and axons being different cell extensions, the mitochondrial transport they mediate may exhibit significant similarities. Furthermore, we found that microtubules may improve the stability and lifespan of B-lymphoma-cell NTs, while F-actin strengthens NTs by providing a structural framework for them. Our results may contribute to a better understanding of the regulation of the major cells of humoral immune response to infections.

## 1. Introduction

The communication between immune cells and the T cell-dependent immune response is typically mediated by a nanoscale gap, known as the immunological synapse between antigen-presenting cells (APCs, such as B lymphocytes, macrophages, dendritic cells (DCs)) and T lymphocytes. In 2004, a distinct form of intercellular communication was identified among immune cells the membrane nanotubes (NTs) [1,2].

NTs are actin-driven membrane protrusions providing physical connection between spatially separated cells even over long distances. Since their first discovery, NTs have been revealed in various mammalian cell types both in vitro, e.g., in neuronal cells, different types of tumour cells, immune cells such as T and B lymphocytes, NK-cells, macrophages, and monocytes [1,3,4,5,6,7,8,9] and in vivo, e.g., in embryonic tissues [10,11,12]. Furthermore, NTs were observed in complex 3D meshworks of different types of tumour cells, such as cell spheroids [13,14]. NTs have been described among atypical lymphocytes in isolated pleural fluid, moreover, from surgically resected and sectioned tumour tissues such as mesothelioma, lung adenocarcinoma, ovarian cancer, and mouse brain [5,15,16,17,18].

NTs show similarity concerning their diameter (100–1000 nm), length (even longer than 100 µm), and cytoskeletal compositions to other membrane projections such as filopodia or cytonemes [19,20]. Despite these similarities, NTs can be easily distinguished from other cell extensions: (i) NTs are thin and straight structures hovering above the substratum in a stretched state between two or more distant cells; (ii) NTs are covered by a continuous membrane and always contain F-actin; furthermore, (iii) they may transfer a wide range of cargos [2,5].

Actin gives the main skeleton of NTs, and it is crucial for their biogenesis. In certain instances, microtubules or intermediate filaments are also identified within the tubes, improving their lifetime and probably stability [21,22,23,24,25,26]. In macrophages, Önfelt and coworkers have described two types of NTs based on their diameter and cytoskeletal composition: thin NTs (diameter (d) < 0.7 µm) that contain only actin and thick ones (d > 0.7 µm) that are also enriched with microtubules [6].

NTs function as special communication nanochannels that facilitate the intensive transport of substances and information between interconnected cells. NTs are conduits for the transportation of various cell organelles, including the microvesicles, endosomes, lysosomes, mitochondria, Golgi apparatus, endoplasmic reticulum (ER), and nucleus [2,3,5,18,27,28,29]. Additionally, different membrane proteins such as CD80 and CD86, as well as genetic information in the form of miRNA, RNA, and DNA, can be delivered through them [5,8,17,30]. Many bacteria, e.g., *Mycobacterium bovis* and *Chlamydia trachomatis* [6,31] and viruses, e.g., HIV [27,32,33,34,35], influenza A [36], human T-cell leukaemia virus type 1/HTLV-1/ [37], and SARS-CoV-2 [38], also use NTs for their intercellular spreading, which enables viruses to remain concealed from the immune system and evade the T cell-dependent immune response.

Mitochondrial dysfunction is suggested to have a role in the deterioration of certain serious neurological disorders [39,40,41], which brought the study of mitochondrial transport into the focus of NT research [6,42]. Mitochondria serve as the primary powerhouses of living organisms and play a crucial role in regulating cell survival or cell death. The dysfunction of mitochondria leads to the accumulation of various reactive oxygen species (ROS), including NADPH oxidase and hydrogen peroxide (H_2_O_2_), which are widely recognised triggers for NT formation [43]. Some studies suggest that mitochondria are transported through NTs that contain microtubules [24,44,45], but mitochondrial delivery has also been observed via NTs that contain only actin [46], indicating that this question has remained unclear.

The diverse functions and structural variations imply that NTs could potentially be formed through various mechanisms. There are two widely accepted hypotheses of NT inception [47]. One of them supports that NTs originate from the initial formation of dorsal filopodia through actin elongation (’making contact’), in which the protrusion comes into physical contact with the surface of a distant cell, resulting in a ‘close-ended’ NT, which, through membrane fusion, may create a continuous channel for the cytoplasm of the cells involved, creating an open-ended tube. The alternative hypothesis suggests that NTs form through cell dislodgement after close contact of two cells (’keeping contact’), resulting in an open-ended, so-called tunnelling nanotube (TNT), which is commonly observed in mobile cells such as immune cells [2,48,49,50,51,52].

In the last few years, we widely studied the TNTs of immune cells, including B-lymphoma cells [5,30,53,54,55]. We found that the composition of TNTs generated by B-lymphoma cells differs from those produced by T cells but exhibits notable similarities to that of the TNTs found in macrophages [6]. While T cells generate a single type of TNTs that contain exclusively actin [9], B-lymphoma cells exhibit two distinct types of TNTs. Approximately 30% of B-lymphoma cells’ TNTs consist solely of actin, while the remaining 70% contain both actin and microtubules. While actin proved to be essential both in the formation and function of these TNTs, the functional significance of microtubules remained unanswered [5,30,56].

Here, we investigated the possible role and localisation of microtubules in the TNTs of B-lymphoma cells. Selective and specific inhibition together with gene silencing of different cytoskeletal elements revealed that microtubules improve the stability and lifetime of B-lymphoma-cell TNTs and mediate mitochondrial transport through kinesin motor proteins, which is also supported by myosin VI. Furthermore, while microtubules are localised mainly at the central axis of the TNTs, actin is physically separated beneath the membrane of the protrusions. The study of the biogenesis of B-lymphoma-cell TNTs revealed that the arrangement and orientation of actin and microtubules are quite diverse, presumably contributing to the diverse function of TNTs connecting B-lymphoma cells. Here we present the initial comprehensive exploration of the potential motor proteins that regulate mitochondrial transport within tunnelling membrane nanotubes. We also show evidence of the resemblance of mitochondrial transport between nanotubes of B-lymphoma cells with specialised functions and axons, which has not been revealed in previous publications.

## 2. Materials and Methods

### 2.1. Cell Culture

B-lymphoma (A20-ATCC, B cells, TIB208, RRID:CVCL_1940) cells of murine origin were cultured in RPMI-1640 medium (Pan-Biotech, Aidenbach, Germany) supplemented with 2 mM ultraglutamine-1 (Lonza, Basel, Switzerland), 1 mM sodium pyruvate (Lonza, Basel, Switzerland), 9.8 µM HEPES (Sigma Aldrich, St. Louis, MO, USA), 50 µM 2-mercaptoethanol (Sigma Aldrich, St. Louis, MO, USA), and 10% FBS (Sigma Aldrich, St. Louis, MO, USA). Cells were maintained in a 25 cm^2^ flask (VWR, Inc., Radnor, PA, USA) at 37 °C, 5% CO_2_ atmosphere and regularly checked for mycoplasma. Fibronectin (Sigma Aldrich, St. Louis, MO, USA) was used to model the optimal extracellular matrix for B-lymphoma cells during cell imaging.

### 2.2. Fluorescent Labelling

To study mitochondria transport, live cells were labelled with fluorescently conjugated MitoTracker Orange CMTMRos dye (50 nM) at 37 °C, 5% CO_2_ atmosphere for 30 min (Thermo Fisher Scientific, Waltham, MA, USA). Silicon rhodamine (SiR) tubulin/actin (Spirochrome Ltd., Stein am Rhein, Switzerland) and Abberior Live 510 tubulin (Abberior GmbH, Göttingen, Germany) fluorogenic probes were used at 1 µM concentration with the addition of 20 µM verapamil (an efflux pump inhibitor promoting the uptake of the dye; included in the kit, Spirochrome Ltd., Stein am Rhein, Switzerland) to label microtubules and actin (37 °C, 5% CO_2_, 1 h long incubation) for live-cell experiments with shorter duration.

To visualise the localisation of mitochondria with microtubules, kinesin, dynein, and myosin II, V and VI motor proteins in the NTs, the cells were fixed with 4% paraformaldehyde (PFA, Sigma Aldrich, St. Louis, MO, USA) for 10 min at room temperature (RT) and permeabilised with 0.2% Tween 20 + 5% bovine serum albumin (BSA) (both from Sigma Aldrich, St. Louis, MO, USA) for 20 min at RT and then incubated separately with monoclonal/polyclonal primary antibodies such as: anti-α-tubulin (1:500; RRID:AB_306044, Abcam, Cambridge, UK), anti-dynein (RRID:AB_668849, Santa Cruz, CA, USA), anti-KIF5B (RRID:AB_2715530, Abcam, Cambridge, UK), anti-myosin II (RRID:AB_291638, BioLegend, San Diego, CA, USA), anti-myosin Va (RRID:AB_649789, Santa Cruz, CA, USA), and anti-myosin VI (catalog number: sc-393558, Santa Cruz, CA, USA), all at a dilution of 1:500, for 1 h. Afterwards they were labelled with secondary antibodies (Alexa488 goat anti-rabbit, RRID: AB_143165/anti-mouse, RRID:AB_2633275; 1:1000, Thermo Fisher Scientific, Waltham, MA, USA) for 45 min at RT and finally mounted in Vectashield mounting medium (Vector Laboratories, Burlingame, CA, USA).

To reveal the localisation of actin with microtubules in the TNTs, the cells were labelled with CF568 phalloidin (Biotium, Fremont, CA, USA), anti-α-tubulin primary (dilution: 1:500; RRID:AB_306044, Abcam, Cambridge, UK), and Abberior STAR RED, as well as goat anti-rabbit secondary (RRID:AB_2833015; Abberior GmbH, Göttingen, Germany) antibodies after the fixation and permeabilisation procedure described above.

### 2.3. Inhibitors

Specific inhibitors were used to reveal the possible role of different cytoskeletal elements in mitochondrial transport via TNTs by selectively blocking the polymerisation of microtubules, as well as the ATPase activity of actin- and microtubule-based motor proteins, preventing their binding to the cytoskeletal polymers and thus hindering their normal functioning. Microtubule polymerisation was blocked with 10 or 20 µM of nocodazole (Thermo Fisher Scientific, Waltham, MA, USA), the activity of dynein was blocked with 20 µM of ciliobrevin D (Sigma Aldrich, St. Louis, MO, USA), and kinesin was inhibited with 15 µM of ispinesib (Selleck Chemicals GmbH, Cologne, Germany). The actin-based myosin II motor protein was blocked with 25 µM of para-nitroblebbistatin (Optopharma Ltd., Budapest, Hungary), myosin V with 30 µM of MyoVin-1 (Merk Millipore, Burlington, MA, USA), and the activity of myosin VI was inhibited with 30 µM of 2,4,6-triiodophenol (TIP) (Alfa Aesar, Haverhill, MA, USA). The optimal concentrations for all inhibitors were determined before each experiment. All dyes, inhibitors, or DMSO (Sigma Aldrich, St. Louis, MO, USA) for control experiments were diluted in culture medium.

### 2.4. siRNAs, Plasmids, and Electroporation

To confirm the involvement of kinesin (KIF5B) and myosin VI motor proteins in the mitochondria transport through TNTs, fluorescently labelled small interfering RNAs (siRNAs) were used (Qiagen, Hilden, Germany): KIF5B siRNA#3 (target sequence: 5′-CAGCAAGAAGTAGACCGGATA-3′, SI02687412), KIF5B siRNA#4 (target sequence: 5′-AACACGAGCTCACGGTTATGC-3′, SI02733437), MYO VI siRNA#3 (target sequence: 5′-CAAGTTCAAGACACAATTAAA-3′, SI01322272), and MYO VI siRNA#4 (target sequence: 5′-CAGCAGGAGATTGACATGAAA-3′, SI01322272), all conjugated with AlexaFluor 488 (AF488) at the 3′ end. Primer sequences for siRNA knockdown are summarised in Appendix A. AF488 AllStars Negative Control siRNA (Qiagen, Hilden, Germany, 1027292), AllStars Mm/Rn Cell Death Control (Qiagen, Hilden, Germany, 1027415), and PBS were used in the negative control, positive control, and mock control experiments, respectively.

LifeAct-RFP (Ibidi GmbH, Gräfelfing, Germany) and -GFP (kind gift from Ferenc Lengyel, Medical School, University of Pécs, Institute of Physiology) plasmids were used to examine the actin polarity in TNTs and, in some cases, the biogenesis of the tubes. EGFP-Tubulin-6 (Addgene plasmid #56450; http://n2t.net/addgene:56450 (accessed on 12 March 2024); RRID:Addgene_56450) [57] and mTagRFP-T-Tubulin-6 were a gift from Michael Davidson (Addgene plasmid #58026; http://n2t.net/addgene:58026 (accessed on 12 March 2024); RRID:Addgene_58026) and were applied to study the microtubule orientation.

siRNAs and plasmids were delivered into the cells by electroporation with an Amaxa 4D Nucleofector system using SF Cell Line KIT L (Lonza, Basel, Switzerland) according to the manufacturer’s instructions (working concentrations: 300 nM/siRNAs and 500–600 ng/µL/plasmids, respectively). The effect of the siRNAs on mitochondrial transport was examined only on cells identified as positive for the green fluorescent signal 72 h post-transfection. To examine the orientation of the cytoskeletal filaments, LifeAct-RFP/GFP (actin) and EGFP/RFP Tubulin-6 (microtubules) transfected cells were mixed in a 1-to-1 ratio and then visualised 24 h later under live-cell conditions.

### 2.5. Microscopic Imaging

Environmental parameters (37 °C, 5% CO_2_ atmosphere) were controlled during all live-cell experiments. The transport of mitochondria and the formation of TNTs were followed by a Zeiss LSM 710 laser scanning confocal microscope (CLSM, 63× magnification, immersion oil objective, NA.: 1.4). The biogenesis of TNTs was followed for 6–12 h under live-cell conditions. Morphological changes in TNTs were visualised at 40× magnification with widefield settings on an Elyra S1 SIM microscope.

Zeiss Elyra S1 superresolution SR-SIM microscopy was used to study the localisation of mitochondria and different motor proteins in TNTs (63× magnification, NA.: 1.4, immersion oil objective, 5 grid rotations) (Carl Zeiss, Oberkochen, Germany). A STEDYCON nanoscope at 100× magnification (oil immersion, NA.: 1.4) was applied (Abberior GmbH, Göttingen, Germany) to reveal the positional relation of the primary cytoskeletal elements in TNTs.

Time-lapse recordings were used to track the movement of mitochondria (40 cycles with a scanning time of 8 s). All treatments were performed for a maximum of 30–40 min to exclude the phototoxic and nonspecific effects of inhibitors. Treatment times for each inhibition are indicated in the relevant parts of the results.

### 2.6. Quantitative Measurements and Statistical Analysis

The calculation of transport velocities and the tracking of mitochondrial motion were carried out with Imaris 8.2 software (Imaris for Cell Biologists, Bitplane, Zürich, Switzerland, RRID: SCR_007370); for velocity calculation, only independent mitochondria were tracked, and a minimum of 35 mitochondria were analysed from at least three independent experiments. Fiji (version 2.14.0, Wayne Rashband, NIH, Washington, DC, USA (RRID: SCR_002285), https://fiji.sc/ (accessed on 13 January 2024)) was used to manually measure the distance travelled by mitochondria over time from their original point.

To assess the effect of inhibitors on the morphology of TNTs, the length and thickness of at least 40 TNTs were measured in Fiji (version 2.14.0, Wayne Rashband, NIH, Washington, DC, USA (RRID: SCR_002285), https://fiji.sc/ (accessed on 13 January 2024)), as published before [54,55]. Circularity and changes in the shape of the cells were determined with Imaris using the following formula f_circ_ = $\frac{4\Pi A}{P^{2}}$, where ‘A’ is the area and ‘P’ is the perimeter of the cell; at least 80 cells were analysed, and all cells were measured separately. Deconvolution was performed in Zen Blue 2.3 software using the constrained iterative algorithm.

Statistical calculations were conducted using Origin 2020 (OriginLab Corporation Northampton, MA, USA—RRID: SCR_014212) or IBM SPSS Statistics (version 26, IBM SPSS Statistics, Armonk, NY, USA—RRID: SCR_019096) software packages. If the Kolmogorov–Smirnov normality test result indicated non-normal distribution, post-hoc tests using Kruskal–Wallis (or ANOVA tests in cases of normal distribution) were performed to compare each treatment group with the control. The Mann–Whitney U test (or Student’s *t*-test in cases of normal distribution) was utilised to prevent type II errors and confirm the significance suggested by the Kruskal–Wallis analysis. A significance level of *p* < 0.05 was set. Data of the statistical analysis are presented as mean ± SD.

## 3. Results

### 3.1. Kinesin and Myosin VI Have a Crucial Role in the Mediation of Bidirectional Mitochondria Transport via TNTs

Both uni- and bidirectional transport of mitochondria were observed; however, the movement of mitochondria through TNTs was not continuous but rather saltatory with frequent stopping phases (Figure 1 and Appendix A).

To identify molecular processes in the background of the observed irregular motion, selective and specific inhibitors of different motor proteins were applied. Nocodazole, a microtubule depolymerising agent, was used at the concentration of 10 µM to follow mitochondrial movement in TNTs. At a higher concentration (20 µM), the mitochondrial movements became immediately faster, and, in most cases, the mitochondria left the TNTs (Appendix A); therefore, for further experiments, a reduced concentration of the agent (10 µM) was used, which seemed to be much more applicable, as mitochondria remained in the TNTs.

The velocity of mitochondrial transport significantly decreased after the addition of the inhibitor (control: 28.16 ± 11.42 nm/s vs. nocodazole treated: 21.06 ± 7.65 nm/s), suggesting that microtubules are involved in the mediation of the transport of mitochondria through B-lymphoma-cell TNTs. Consequently, cells were subjected to specific and selective inhibitors of kinesin and dynein proteins to identify microtubule-based motor proteins associated with this mitochondrial transport. While the average velocity of mitochondria in TNTs was significantly reduced due to inhibition of the kinesin activity (20.86 ± 12.89 nm/s), the inhibition of dynein had an opposite effect and somewhat increased the transport rate (33.51 ± 18.42 nm/s). The lack of complete inhibition suggested the potential involvement of other motor proteins that facilitate the bidirectional transport, enabling mitochondria to be transported in the opposite direction through TNTs. To explore this possibility, the activity of myosin II, V, and VI, motor proteins reported to be typically involved in actin-based transport processes, were also blocked [45,58,59]. The inhibition of myosin VI significantly reduced the transport velocity (23.19 ± 14.17 nm/s). In contrast, selective blocking of myosin II and V, similar to the effect of dynein inhibition, slightly enhanced the movement of mitochondria in the TNTs (31.89 ± 12.43 nm/s and 33.63 ± 22.28 nm/s, respectively). The co-inhibition of kinesin and myosin VI motor proteins resulted in a more pronounced decrease in the transport rate (12.93 ± 4.82 nm/s), confirming their significant contribution to the delivery of mitochondria (Figure 2 and Figure 4A,C and Appendix A).

To verify the results of the inhibition experiments, siRNAs were used to repress the expression of kinesin and myosin VI. Because A20 cells are not ideal for transfection, siRNAs were conjugated with a fluorescent signal to assess transfection efficiency. Only TNTs that exhibited the fluorescent signal of siRNA were analysed. Statistical analysis of live-cell microscopic data revealed that the depletion of kinesin and myosin VI expression significantly decreased mitochondrial mobility (negative control: 36.27 ± 20.59 nm/s, KIF5B siRNA#3: 23.58 ± 12.62 nm/s, KIF5B siRNA#4: 17.11 ± 6.62 nm/s, MYO VI siRNA#3: 23.57 ± 13.47 nm/s, MYO VI siRNA#4: 18.21 ± 8.15 nm/s (Figure 3 and Figure 4B,D and Appendix A).

Furthermore, we examined the localisation of mitochondria and motor proteins in TNTs. Superresolution micrographs revealed a positional relation of mitochondria with microtubules, kinesin, and myosin VI (Figure 5).

### 3.2. Microtubules May Improve the Stability of B-Lymphoma-Cell TNTs

Here, we demonstrated that microtubules play a role in facilitating transport processes via B-cell lymphoma TNTs. Furthermore, a significant change in the morphology of TNTs was observed upon the treatment of cells with the microtubule depolymerising agent nocodazole (Figure 6). Nocodazole at a larger (20 µM) concentration cannot be applied because it caused 27% of the TNTs to break, making the analysis impossible. Due to the application of a lower concentration (10 µM) of nocodazole, only 13% of the TNTs tore (Figure 6A), and their diameter was significantly reduced after only 5 min of treatment, which became more pronounced with an increasing duration of incubation in the presence of nocodazole (Figure 6B,C and Table 1). On the other hand, nocodazole treatment with different time periods did not affect the length of TNTs (Figure 6D and Table 1) or the shape of B-lymphoma cells (Figure 6E and Table 1), suggesting that the microtubular network plays a crucial role not only in some transport processes but also in the mechanical stability and lifetime of TNTs, consistent with previous publications [6,45,51,60].

The arrangement of the main cytoskeletal elements of B-lymphoma-cell TNTs further supports that the presence of microtubules may contribute to the stability of TNTs. STED imaging revealed that while actin filaments form a kind of lining that is located mainly beneath the membrane of TNTs, and microtubules are spatially separated and primarily exhibit central localisation along the entire length of the tubes (Figure 7).

### 3.3. Cytoskeletal Distribution May Promote Bidirectional Transport Processes within B-Lymphoma-Cell TNTs

Although the cytoskeletal composition of B-lymphoma-cell TNTs was studied earlier [5], the specific details of the localisation, development, and orientation of the cytoskeletal elements in these TNTs are still unknown. Furthermore, the significance of these features in the biological function of nanotubes is yet to be fully understood. Therefore, to reveal whether the arrangement of the cytoskeletal filament systems contributes to bidirectional transport processes, we fluorescently labelled actin and microtubules in B-lymphoma cells, then followed the biogenesis of TNTs with live-cell CLSM. The images proved that TNT induction of B-lymphoma cells requires direct cell–cell interaction; subsequently, the tubules elongate as the previously contacted cells displace or detach from each other, eventually creating an open-ended TNT (as suggested before [5,30] (Figure 8)). Roughly ~77% of the observed membrane tethers were formed by the mechanism just described, while ~23% developed during cell division, where a parent cell remained connected to its progeny through a bridge at the end of the cytokinesis, similarly to T cells [9]. We note here that the intercellular bridges formed during incomplete cytokinesis were not considered nanotubes (remaining consistent with the literature [9].

Based on our observations, the cytoskeletal composition of TNTs influenced their mechanical stability and lifetime. Thin TNTs that were composed solely of filamentous actin (with no detected microtubules) appeared to be more delicate and exhibited a shorter lifespan compared to the tubes that also contained microtubules and had a greater diameter. Moreover, microtubules in thicker tubes were present even in the early phase of their initiation (Figure 8).

Based on our microscopic analysis, the actin pattern in B-lymphoma-cell TNTs shows differences, creating two main populations. (1) Tubes may contain actin from only one of the cells (Figure 9A), or (2) tubes may be enriched with actin derived from both contributing cells, resulting in the formation of TNTs where actin filaments of distinct origins are either segregated (Figure 9B) or partially overlap (Figure 9C and Appendix A).

Similarly to actin, microtubules also show different patterns in the examined TNTs, thus creating three groups of nanotubular projections. (1) Some TNTs contained microtubules in their entire length that originated only from one of the cells participating in the tube formation (Figure 10A). (2) In certain cases, although microtubules grew into the TNT from a given direction, they occupied only a short section of the TNT and were not observable in the tube’s whole length (Figure 10B). (3) In other tubes, while microtubules were present along the whole axis of the tube, they were grown from the direction of both contributing cells (Figure 10C). Interestingly, when microtubules grew unidirectionally into the TNTs (i.e., in group (1) and (2)), the microtubule network of the receding cell was involved in the formation of the protrusion.

## 4. Discussion

The discovery of membrane nanotubes has opened new pathways in understanding intercellular communication. NT networks both in prokaryotes and eukaryotes play a crucial role in facilitating material transport in various cell types both in vitro and in vivo, influencing several developmental processes and the course of some diseases [56,61]. Based on the results of numerous studies, it is known that NTs exhibit heterogeneity in their structural characteristics such as length, diameter, and lifetime and in their formation, cytoskeletal composition, and mediated transport mechanisms. Nonetheless, the functional importance of NTs in antigen-presenting cells and B cells, which play a vital role in the defence against pathogens, has not been extensively studied.

Based on our previous and current findings, although B-lymphoma cells are specialised components of the immune system, their TNTs exhibit similarities to those observed in other cell types, not only in their function but also in terms of cytoskeletal composition. For instance, the arrangement of the main cytoskeletal filaments in B-lymphoma-cell TNTs is possibly very similar to that observed in macrophages [6]. Moreover, TNTs with similar actin orientation were observed between HeLa cells [62,63] suggesting that this is probably not a cell-specific but a general phenomenon, regardless of the fact that transport processes via NTs may be mediated by different molecular mechanisms for each cell and that the functions of different cells may vary significantly.

Consistent with findings in other cells, our results demonstrated that the primary cytoskeletal elements have multiple functions in the TNTs of B-lymphoma cells. Actin is essential for their formation. Additionally, it plays a vital role in establishing the framework for diverse transport processes; the heterogeneous orientation of actin filaments facilitates (bidirectional) microvesicular transport between B-lymphoma cells [5,30] and potentially contributes to the elasticity of NTs [64]. While our previous study demonstrated that microtubules are not required for the initiation of TNTs [5], our current results indicate that the presence of the microtubular network is crucial in B-lymphoma-cell TNTs, albeit in the later phase of their life cycle. Namely, microtubules not only facilitate mitochondria delivery through the kinesin motor but also improve the stability of the tubes, leading to longer lifetimes and more intense transport processes. It is supported by the fact that microtubules exhibit a stiffness that is three orders of magnitude greater than that of actin [60] and possess a significant persistence length, allowing them to retain their shape and anisotropy [65]. This results in a longer lifetime of TNTs containing microtubules compared to those containing only actin [66]. Furthermore, the force generated along cell protrusions (at least in axons) provokes the stabilisation of the microtubules within these protrusions, leading to their decreased turnover. This, in turn, facilitates their accumulation and promotes the transport processes mediated by them [24,67]. This is particularly notable as T lymphocytes, which lack microtubules entirely, do not exhibit similar transport characteristics [9]. It is worth noting that in B-lymphoma cells, TNTs display remarkable elasticity, as evidenced by their impressive resilience to repeated high-amplitude stretches. They swiftly return to their original shape after stretching, suggesting a function akin to elastic cords in response to environmental tension [53]. This observation becomes intriguing when considering that most of the B-lymphoma-cell TNTs contain microtubules beyond actin [5] and the growing end of microtubules under tension tends to elongate [68,69], potentially influencing (de facto decreasing) microtubule depolymerisation rates [65], thus contributing to the mechanical stability. Our findings support that by providing structural support and regulating TNT dynamics, the cytoskeleton ensures the proper functioning and durability of these cellular protrusions. Nevertheless, a broader investigation should be conducted to study the mechanical properties of microtubule-stabilised TNTs in detail.

Due to its involvement in neurological diseases, mitochondrial transport is one of the most widely studied processes in NT research. Based on these investigations, the direction of mitochondria transport may vary, and depending on the cell type, it can be either unidirectional, where mitochondria are transported from healthy cells to damaged cells via NTs supporting cell survival (e.g., from pericytes to astrocytes, in PC12 and normal porcine urothelial /NPU/ cells, or from bone marrow stromal cells to acute myeloid leukaemic cells to improve resistance against chemotherapy) [24,45,70,71] or bidirectional (e.g., between human umbilical vein endothelial cells /HUVECs/ and mesenchymal stem cells /MSCs/, or between Jurkat cells and MSCs [72,73,74,75,76]. The main molecular components of mitochondrial delivery have already been identified; however, since these results are often contradictory, the exact mechanism of their transfer is still unclear. In addition, regulatory proteins involved in the mitochondrial transport of B cell TNTs were not investigated at all. Several studies have confirmed that the movement of mitochondria can be facilitated by various motor proteins that rely on the cytoskeleton. Certain studies linked the transport of mitochondria to the microtubule-based kinesin motor protein [24,44,72]. Others propose that the mitochondrial transport in NTs may occur independently of microtubules, suggesting an alternative mechanism [46,77]. In neurons, microtubule-based motor proteins play a crucial role in regulating the movement of mitochondria, kinesin is responsible for facilitating anterograde transport, and dynein controls retrograde movements [78,79]. Microtubules were reported to regulate the movement of mitochondria over long distances in axons, and simultaneously, the actin network proved to be crucial for anchoring mitochondria and facilitating their movement over shorter distances [79,80]. Other studies conducted on neurons have explored the interaction between mitochondria and various myosins (II, V, VI, and XIX). These findings revealed that the downregulation of myosin V and VI led to a notable increase in the velocity of mitochondrial transport, suggesting that these two motor proteins are likely responsible for the docking of mitochondria along actin [59,79,81], which may play an important role in neurons in the proper distribution of mitochondria along long axons, which is required for local energy supply in regions with high energy consumption [82].

The results of our inhibitory and gene silencing experiments are in concert with those obtained in neurons, indicating the involvement of both microtubule- and actin-based motor proteins in the transportation of mitochondria through B-lymphoma-cell TNTs, suggesting that mitochondrial movement towards the (+) end of the microtubule is regulated by kinesin, while myosin VI is responsible for the movement in the opposite direction along actin. Increased velocity in mitochondrial transport observed after the treatment with dynein, myosin II, and V blockers may be the result of the stress caused by the addition of the inhibitors, or these molecular motors may facilitate the docking of mitochondria along the microtubule or the actin network, as was suggested in neurons [59,79], contributing to the discontinuous movement (active motion alternating with pause phases) of mitochondria in TNTs. The blocking of these proteins disturbs their function, preventing them from anchoring mitochondria to the cytoskeleton resulting in increased transport velocity with a modified trajectory and more uniform movement [59]. Our hypothesis is supported by our superresolution SIM images, where mitochondria showed co-localisation with microtubules, kinesin, and myosin VI, which are present in abundance near the mitochondria, similarly to dynein and sometimes myosin II and V (Appendix A). It is important to note that both actin- and microtubule-based motor proteins are able to bind to the same cargo, resulting in the cargo switching between actin and microtubules during its transportation. Furthermore, due to electrostatic interactions, microtubule-based motors may be connected with actin and myosins with microtubules, at least in vitro in the case of kinesin and myosin V [83], thus contributing to the successful transport of the cargo. As far as we know, our study is the first to extensively investigate the potential motor proteins regulating mitochondrial transport within nanotubes. No prior publications have demonstrated the similarity of mitochondrial transport in nanotubes—particularly those formed among B-lymphoma cells with specialised functions—and axons.

Figure 11 illustrates the schematic representation of our model, which is based on our findings concerning the cytoskeletal elements constituting B-lymphoma-cell nanotubes and their involvement in transport processes.

B cells are essential components in immune defence by producing antibodies in a highly orchestrated action with the cells of the innate immune system, antigen-presenting cells, and various T-cell subsets. Among others, TNTs may act as one of the mechanisms controlling the above orchestration, e.g., through the intercellular transport of important regulatory molecules or organelles [5,30,56,61,85,86] via TNTs between distant immune cells.

Mitochondria are essential energy plants for all the cells of the immune system and can keep them alert through the production of reactive oxygen species (ROS), which may even control cellular differentiation, B cell fate, survival, etc. [87,88]. In addition, recently, it has also been shown that cytosolic escape of mitochondrial DNA may even trigger the activation of innate immunity. Furthermore, the transport of mitochondria from mesenchymal stem cells to cells defective in mitochondrial DNA via TNTs was reported [88]. In the case of infection, mitochondria may also release DAMPs (danger-associated molecular patterns) that in turn can switch on an inflammatory immune response against the pathogens [89]. The intercellular transport of mitochondria via TNTs between various immune cells may help to increase the efficacy of the utilisation of DAMPs. Finally, an interesting recent work [90] reported that MitoMiRS (micro RNAs of mitochondrial origin) may control mitochondrial functions and metabolism alike. Such short, non-coding miRNAs could be transported between cells via TNTs either inside intact mitochondria or as a released macromolecule [56,61]. The miR-17/92 cluster has been shown to regulate mitochondrial metabolism. This cluster is deregulated, among others, in B-cell lymphomas, B-cell chronic lymphocytic leukaemia, acute myeloid leukaemia, and T-cell lymphomas. Thus, we conclude that nanotubular intercellular transport of mitochondria in the immune system may potentially control immune cell activities in the case of infections as well as the formation of lymphoid tumours.

## 5. Conclusions

Our study may contribute to a better understanding of the various aspects related to the molecular mechanism of the transport of mitochondria between immune cells. By utilising TNTs, it may be possible for a cell to transfer mitochondria to another (for instance damaged) cell, thereby potentially restoring the energetic profile of the recipient cell. Gaining insights into the functional significance of TNTs, improving our knowledge about their composition, and elucidating their cytoskeletal orientation enables TNTs to be possible therapeutic targets in the treatment of different disorders associated with mitochondrial dysfunction or origin [91,92].

## Figures and Tables

**Figure 1 cells-13-00607-f001:**
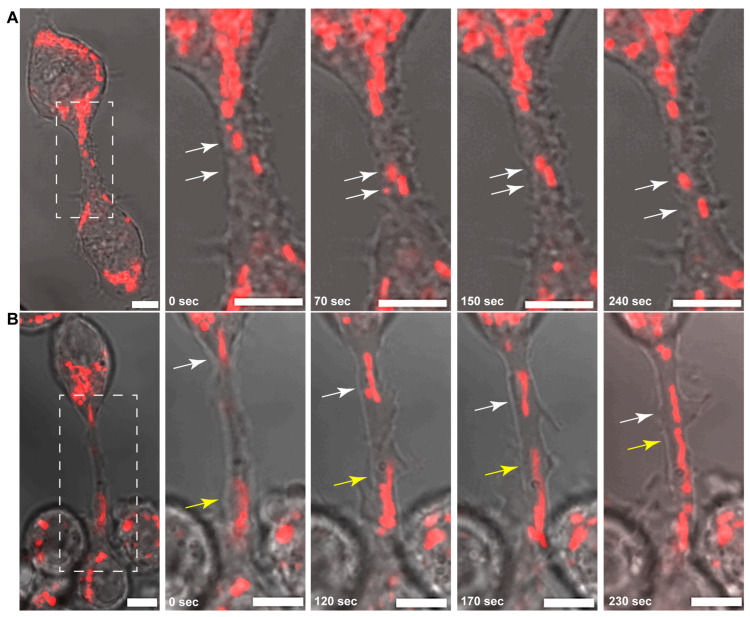
Uni- and bidirectional transport of mitochondria between B-lymphoma-cell tunnelling membrane nanotubes (TNTs). (**A**) Representative live-cell confocal microscopic images of intensive uni- and (**B**) bidirectional movements of fluorescently labelled mitochondria in control conditions. Arrows (white and yellow) depict the position of the mitochondria in the zoomed images of the boxed regions. During unidirectional transport (**A**), mitochondria move via TNTs from the donor cell towards an acceptor one, while in bidirectional movements (**B**), mitochondria originate from both cells that are connected by TNTs and move in opposite directions (i.e., both cells act as both donors and acceptors simultaneously). Scale bars: 5 µm.

**Figure 2 cells-13-00607-f002:**
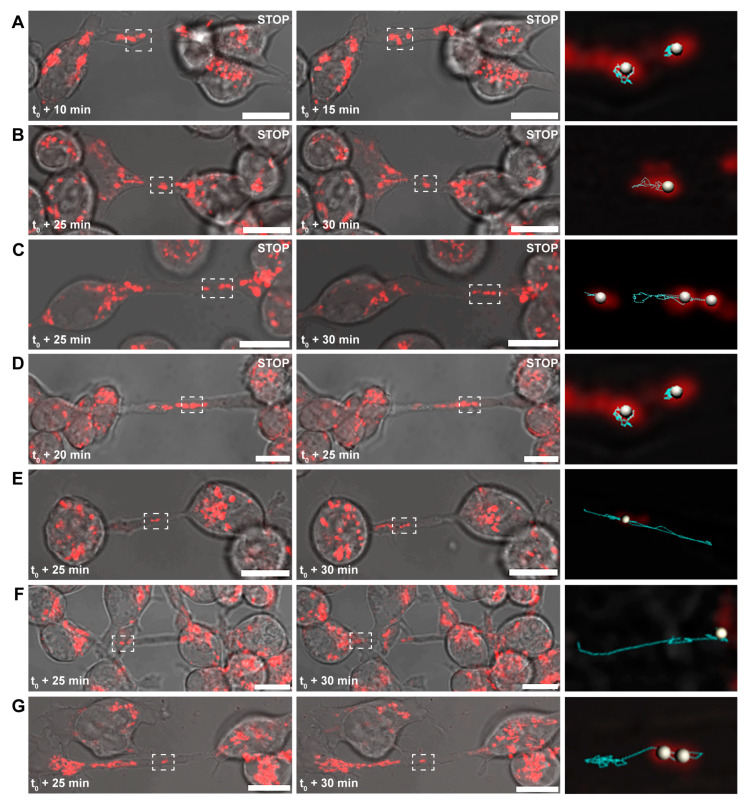
Representative live-cell confocal micrographs of the effect of inhibitors on mitochondrial transport. (**A**) Obstruction of mitochondrial movement due to the disruption of microtubule assembly/disassembly following nocodazole treatment. (**B**) Inhibition of mitochondrial transport due to the loss of the kinesin activity. (**C**) Myosin VI also impedes mitochondria delivery in TNTs. (**D**) Co-inhibition of kinesin and myosin VI motor proteins blocks mitochondrial movements. (**E**–**G**) Blocking of dynein, myosin II, and myosin V activities did not hinder the motion of mitochondria. Motion trajectories of single mitochondria were analysed within the boxed areas (white dots represent the analysed mitochondria; blue lines show their movement pattern); t_0_ indicates the addition of a given inhibitor. As inhibitors require a minimum of 10–20 min to exert an effect on the target, the micrographs depict the states observed after these time intervals. Scale bars: 10 µm.

**Figure 3 cells-13-00607-f003:**
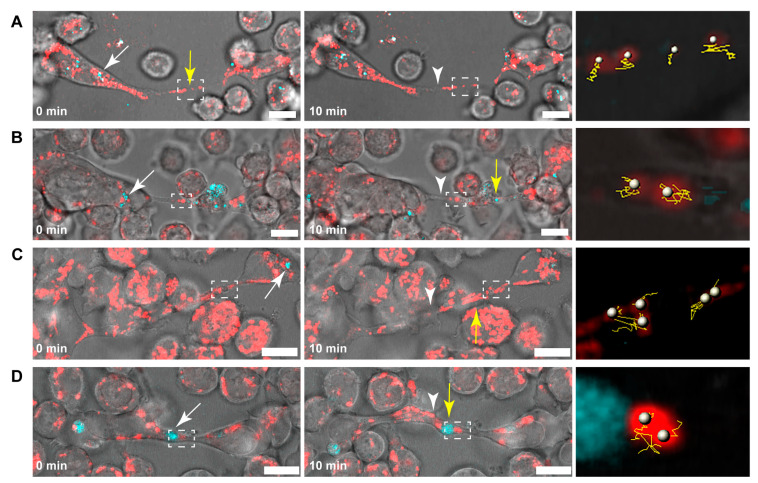
Effect of kinesin and myosin VI silencing on mitochondrial transport in B-lymphoma-cell TNTs. (**A**,**B**) Mitochondrial transport is reduced by KIF5B siRNA#3 and siRNA#4. (**C**,**D**) Depletion of myosin VI expression resulted in shorter motion trajectories. The analysed mitochondria are indicated by the boxed regions. Examples of the fluorescent signals of the siRNA are indicated by white (in the cell) and yellow (in the TNTs) arrows, and arrowheads indicate TNTs. Scale bars: 10 µm.

**Figure 4 cells-13-00607-f004:**
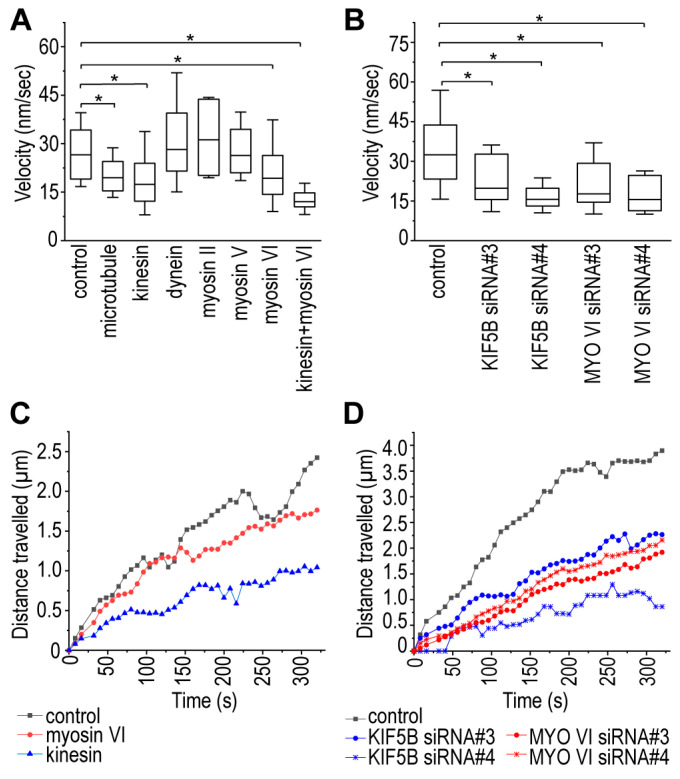
Statistical analysis of the transport velocities and distance travelled by mitochondria in TNTs. (**A**) Change in the transport velocities due to the selective inhibition of the examined cytoskeletal elements. (**B**) Decrease in mitochondrial transport velocity due to gene silencing. Average distance travelled from the starting point by the mitochondria over time due to (**C**) the effect of selective and motor protein specific inhibitors and (**D**) the depleted expression of kinesin and myosin VI proteins; as a negative control, cells were transfected with AF488 AllStars Negative Control siRNA. Results are shown as mean ± SD, *p* < 0.05, indicated with an asterisk. A minimum of 35 mitochondria were analysed from at least three independent experiments in both the control and the treated samples.

**Figure 5 cells-13-00607-f005:**
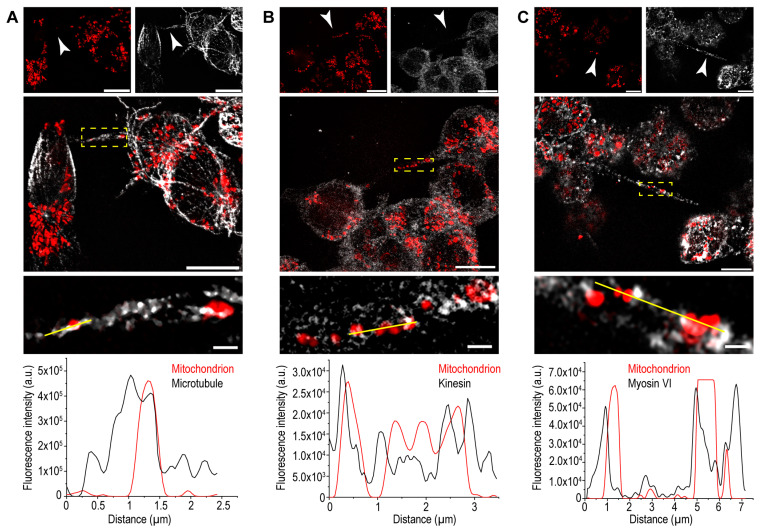
Localisation of mitochondria and motor proteins in B-lymphoma-cell TNTs. (**A**) Representative SR-SIM superresolution images show co-localisation between microtubules and mitochondria, which is supported by the line scan analysis; the analysed region is shown by the yellow line. Red: mitochondria, grey: microtubules. The micrograph at the bottom is a zoomed image of the yellow-boxed area. (**B**,**C**) Both kinesin (**B**) and myosin VI (**C**) show positional relations with mitochondria. The intensity profiles confirm the comparable distribution of the mitochondria with kinesin and myosin VI; the analysed regions are shown by the yellow lines. Red: mitochondria, grey: (**B**) kinesin, (**C**) myosin VI. Arrowheads depict TNTs. Micrographs at the bottom are zoomed from the yellow-boxed areas. Scale bars: 10 µm, and in the zoomed images: 1 µm.

**Figure 6 cells-13-00607-f006:**
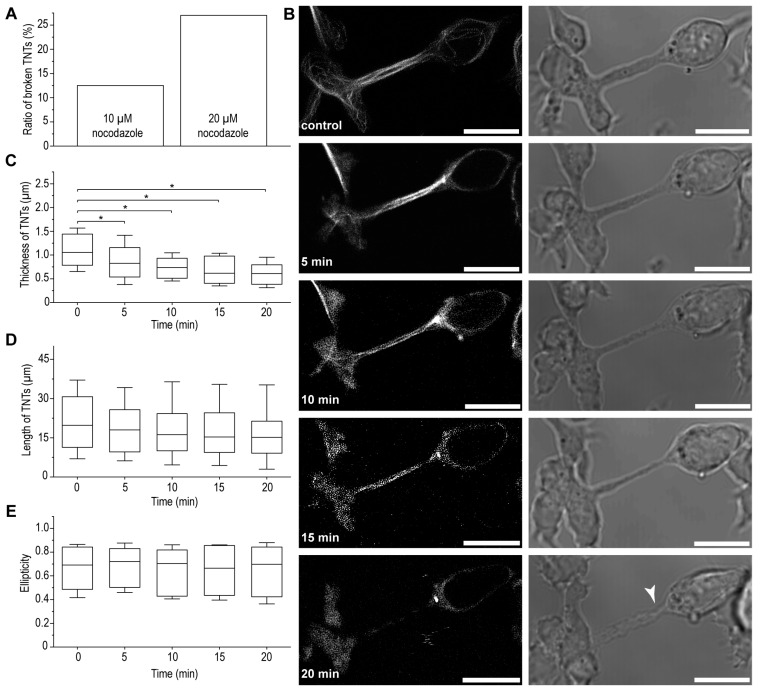
Microtubules improve the stability of B-lymphoma-cell TNTs. (**A**) A 20 min treatment with 20 µM of nocodazole caused ~1/3 of the TNTs to break, while 10 µM proved to be suitable for the examination of TNTs. (**B**) Nocodazole-treated cells showed a fragmented microtubule network (left, SiR tubulin-labelled cells, deconvolved CLSM images) and a decrease in the thickness of TNTs over time (right, WF images, arrowhead indicates a significant reduction in the diameter of the TNT around the growth cone, which is probably a condition before the retraction/rupture of the TNT), scale bars: 10 µm. (**C**) Statistical analysis of our microscopic observations revealed that the thickness of TNTs was significantly reduced 5 min after the addition of the inhibitor. (**D**) Nocodazole treatment did not change the length of the TNTs or (**E**) the shape of the cells. Results are shown as mean ± SD, *p* < 0.05, indicated with an asterisk.

**Figure 7 cells-13-00607-f007:**
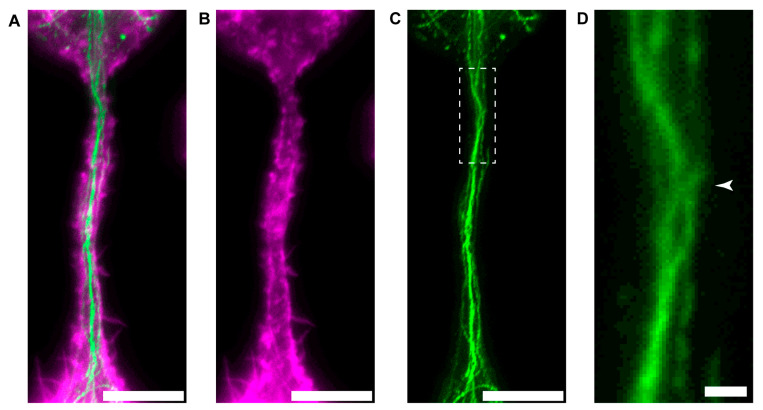
Localisation of the cytoskeletal filaments in B-lymphoma-cell TNTs. (**A**–**C**) The representative STED micrographs display that actin (**B**) is mainly located in the wall of the TNTs, serving as its structural framework and creating a stable tunnel between the cells, and the microtubules (**C**) are longitudinally positioned throughout the lumen of the tube, exhibiting a central location and sometimes showing twisting. Cells were fixed and then labelled for the presence of (**B**) actin (magenta, phalloidin 568) and (**C**) microtubules (anti-α-tubulin primary and Abberior STAR RED goat anti-rabbit secondary antibodies, green). (**A**) Merged image of actin and microtubule. (**D**) Zoomed image of the boxed region of (**C**) showing twisted microtubules (arrowhead). Scale bars: 5 µm (**A**–**C**), 0.5 µm (**D**).

**Figure 8 cells-13-00607-f008:**
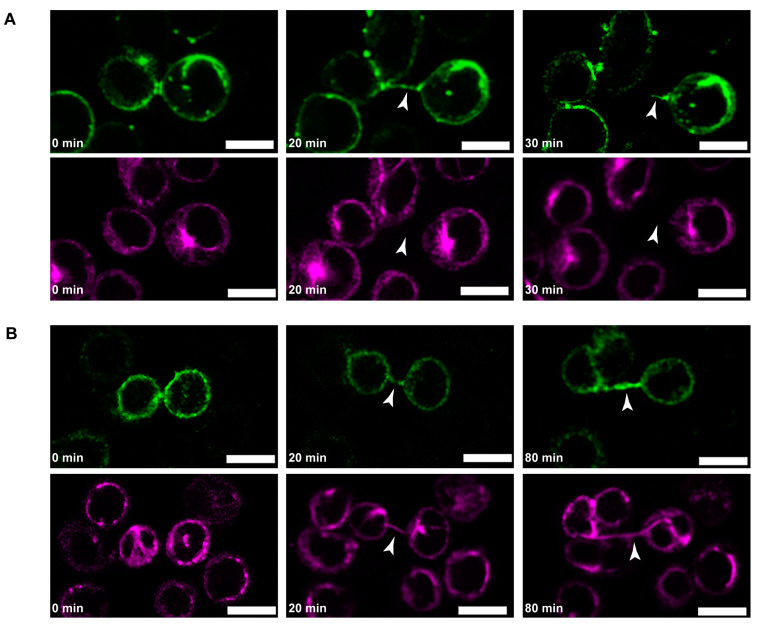
Representative live-cell confocal images of the formation of B-lymphoma-cell TNTs with different cytoskeletal compositions. (**A**) TNTs that formed after cell dislodgement and contain only actin have a shorter lifetime; their diameter decreases over time, and the tubes finally break apart within 20–30 min; green: actin (cells were labelled with SiR actin), magenta: microtubules (cells were labelled with Live 510 tubulin). (**B**) TNTs that are rich in microtubules possessed longer lifespans, but their formation seemed to be slower; green: actin (cells were transfected with LifeAct-RFP), magenta: microtubules (Live 510 tubulin labelled). Arrowheads indicate NTs. Scale bars: 10 µm.

**Figure 9 cells-13-00607-f009:**
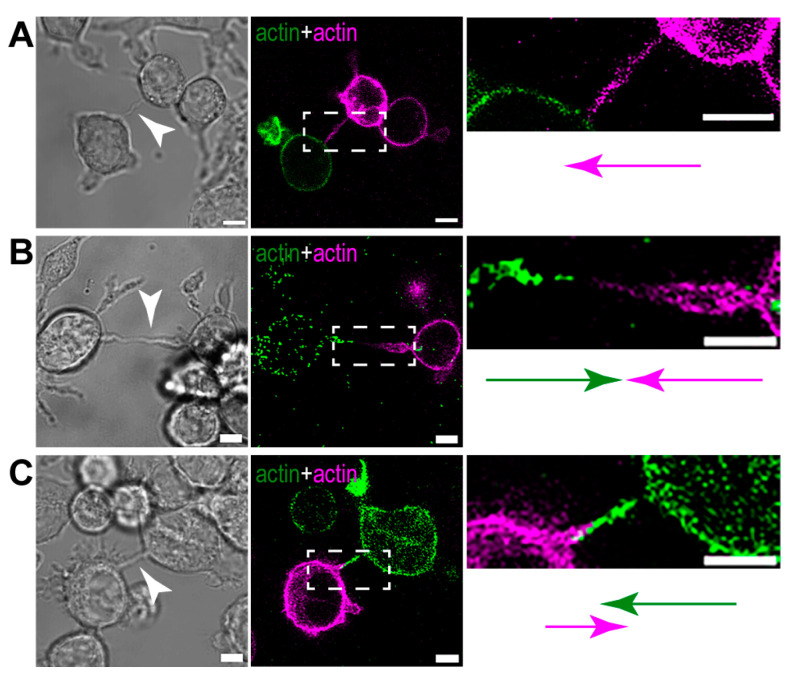
TNTs of B-lymphoma cells exhibit different actin patterns. (**A**) Certain TNTs grow unidirectionally and contain actin of only one of the participating cells (the image on the right is the magnification of the boxed region in the middle image). TNTs can also grow bidirectionally, where the tubes are enriched with actin from both cells, creating two different patterns: actin filaments in the TNTs of the participating cells are either (**B**) completely separated or (**C**) overlap to a certain extent. Images on the right side are zoomed imaged of the appropriate boxed regions. The arrows with different colours (green and magenta) show the directionality of actin bundles originating from distinct cells. Corresponding brightfield images are displayed on the left, and TNTs are depicted by arrowheads. Cells were transfected either with LifeAct GFP or RFP to visualise actin in the TNTs. Scale bars: 5 µm.

**Figure 10 cells-13-00607-f010:**
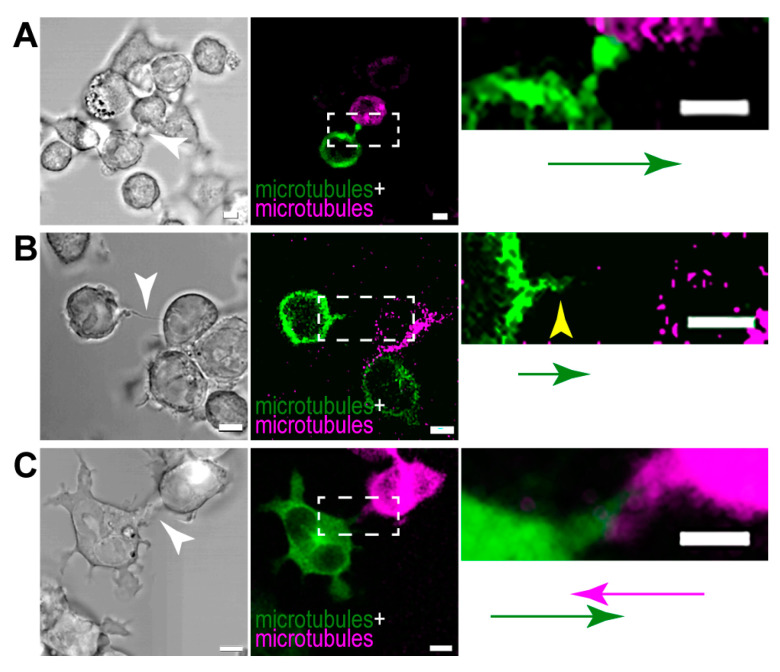
Microtubules show different orientations in B-lymphoma-cell NTs. Representative CLSM images show that microtubules may grow into the TNTs unidirectionally and either (**A**) present along the whole length of the tube or (**B**) develop only to some extent (yellow arrowhead in the right panel). (**C**) TNTs can contain microtubules that originate from both cells participating in the creation of the tube. The arrows (green and magenta) represent the direction of microtubule growth in the TNTs. Images on the right are zoomed images of the appropriate boxed regions. Corresponding brightfield images are displayed on the left, TNTs are depicted by arrowheads. Cells were transfected either by mTag RFP-T-Tubulin-6 (magenta) or EGFP-Tubulin-6 (green) and cocultured in a 1:1 ratio. Scale bars: 5 µm.

**Figure 11 cells-13-00607-f011:**
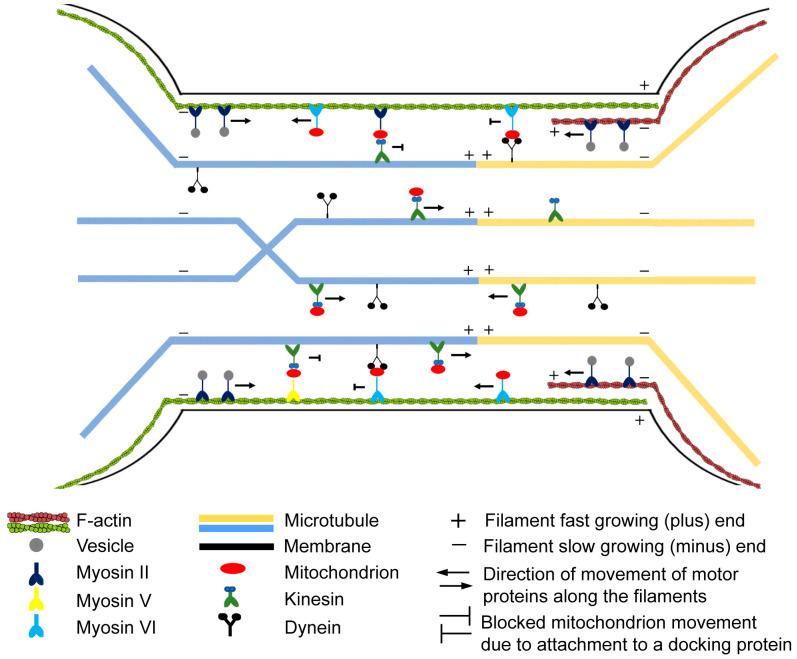
Schematic representation of the cytoskeletal arrangement and function of TNTs between B-lymphoma cells. The dislodgement of the cells leads to the formation of open-ended TNTs, facilitating the transportation of various cargos. The actin filaments are positioned beneath the membrane that covers the TNTs, while the microtubule bundles predominantly run longitudinally along the centre of the TNTs, probably because due to their high persistence length, microtubules exhibit sensitivity to cell geometry and align axially, leading to the arrangement of microtubule networks along straight structural elements [65]. Moreover, microtubules sometimes exhibit a twisted pattern, similar to the TNTs of urothelial cells [84]. Actin and microtubules have the potential to extend into the nanotube from both participating cells involved in its formation. Transport processes occurring within B-lymphoma-cell nanotubes can be facilitated by both actin- and microtubule-mediated mechanisms. Myosin II is responsible for the carriage of microvesicles [30]. The bidirectional movement of mitochondria is accomplished through the collaboration of two motor proteins: kinesin transports the mitochondria towards the plus end of microtubules, whereas myosin VI conveys them in the opposite direction along actin filaments. Mitochondrial movement is discontinuous and characterised by frequent active and stopping (inactive) phases. We suppose that dynein, myosin II, and myosin V motor proteins are responsible for the anchoring or docking of the mitochondria along the cytoskeleton of the TNT, contributing to the stationary phase of the transport. The process of mitochondria transport enters a resting phase upon the binding of motor proteins responsible for transportation and the corresponding docking motor proteins. The continuous movement of mitochondria becomes apparent when the docking motor proteins are released.

**Table 1 cells-13-00607-t001:** Results of the analysis of morphological changes in TNTs and cells due to nocodazole treatment of different durations.

Time (min)	Average Thickness of TNTs (µm)	Average Length of TNTs (µm)	Roundness of Cells
0	1.11 ± 0.457	22.02 ± 15.09	0.64 ± 0.22
5	0.898 ± 0.517	20.22 ± 14.03	0.67 ± 0.21
10	0.75 ± 0.296	20.52 ± 15.90	0.63 ± 0.23
15	0.694 ± 0.344	19.10 ± 15.52	0.63 ± 0.23
20	0.633 ± 0.321	21.27 ± 16.12	0.62 ± 0.26

## Data Availability

The data presented in this study are available on request from the corresponding author.

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
