# Peer review of "Cooperation of Various Cytoskeletal Components Orchestrates Intercellular Spread of Mitochondria between B-Lymphoma Cells through Tunnelling Nanotubes"

_cells, 2024, doi:10.3390/cells13070607_

Round 1

Reviewer 1 Report (Previous Reviewer 1)

Comments and Suggestions for Authors

In the article entitled “Cooperation of Various Cytoskeletal Components Orchestrates Intercellular Spread of Mitochondria Between B-Lymphoma Cells Through Tunnelling Nanotubes” the authors proposed a study on the possible localization and role of microtubules in tunnelling nanotubes of B-Lymphoma cells, presenting also a comprehensive exploration of the motor proteins responsible for mitochondrial transport between different B cells. They demonstrated how the presence of the microtubule network is a key factor in the formation of TNT between B-Lymphoma cells and that the cytoskeleton guarantees appropriate function and prolonged durability to these structures. They carried out an extremely clear and comprehensive investigation, starting from the background provided to the materials and methods used, the results shown, and the extensive discussion of the work done. The references reported are summarily appropriate and include recent literature data, in line with the present study. The authors have also revised the manuscript according to the indications given in the previous revision, making all the suggested corrections

Reviewer 2 Report (Previous Reviewer 2)

Comments and Suggestions for Authors

Acknowledging your previous publications and your arguments that tumour microtubes may stay only in glioblastoma cells (which is debatable), the reviewer must say that the protrusions that you are showing in Figure 1A and B, far larger in diameter than 1 micrometer. Cells do have many kinds of extensions. During cell division and cell body separations, they remain connected via adjacent membranes. Furthermore, there is no evidence that mitochondria are transferring from one cell to another. The tubes can be closed-ended, and the authors have shown only the traveling of mitochondria through these protrusions. The paper can not be accepted. It will give wrong information to the field of TNTs.

Comments on the Quality of English Language

NA

Reviewer 3 Report (New Reviewer)

Comments and Suggestions for Authors In the revised version authors addressed and deeply discussed all my comments and concerns. To support their point of vie, authors provided solid, significant arguments together eith references. Even if I still do not agree with all the arguments, this is still an area of general scientific discussion, so I accept it. This field has still many open questions whic are matter of open debate.  New important experiments and information were added, which improved the manuscript and data quality. Especially control experiments and verification of cytotoxicity of used treatments were necessary. 

This manuscript is a resubmission of an earlier submission. The following is a list of the peer review reports and author responses from that submission.

Round 1

Reviewer 1 Report

Comments and Suggestions for Authors

The authors proposed a study on the possible localization and role of microtubules in tunnelling nanotubes of B lymphocytes, presenting also a comprehensive exploration of the motor proteins responsible for mitochondrial transport between different B cells. They demonstrated how the presence of the microtubule network is a key factor in the formation of TNT between lymphocyte B cells and that the cytoskeleton guarantees appropriate function and prolonged durability to these structures. They carried out an extremely clear and comprehensive investigation, starting from the background provided to the materials and methods used, the results shown, and the extensive discussion of the work done. 

The following are only some suggestions to better illustrate the present work.

-          Line 117, please change cm2 into cm2

-          The involvement of the microtubule network of the receding cell in protrusion formation is reported in Figure 10. However, this does not seem well detectable, particularly in panel B. It might be helpful to indicate it, maybe using arrows or arrowheads, in the zoomed image on the right.

Author Response

Response to Reviewer 1 Comments

We appreciate the time and effort that you have dedicated to provide your valuable feedback on our manuscript. We are grateful for the insightful comments on our paper.

We have been able to incorporate changes to reflect the suggestions provided by Reviewer 1, and we have highlighted the changes within the manuscript.

Here is a point-by-point response to the Reviewer’s comments and concerns.

Point 1: Line 117, please change cm2 into cm2

Response 1: Thank you for pointing this out, we have revised our manuscript accordingly.

Point 2: The involvement of the microtubule network of the receding cell in protrusion formation is reported in Figure 10. However, this does not seem well detectable, particularly in panel B. It might be helpful to indicate it, maybe using arrows or arrowheads, in the zoomed image on the right.

Response 2: Thank you for this suggestion. The microtubule network, originating from only one of the cells that participated in the nanotube formation has been marked with an arrowhead in Figure 10B, zoomed image, and the corresponding figure caption was revised accordingly (page 16, line 424).

We look forward to hear from you in due time regarding our submission and to respond to any further questions and comments you may have.

Reviewer 2 Report

Comments and Suggestions for Authors

I have now read the manuscript carefully. The study “Cooperation of Various Cytoskeletal Components Orchestrates Intercellular Spread of Mitochondria Between B Cells Through Tunnelling Nanotubes” try to distinct the role of actin and tubulin in TNTs and their role in mitochondria transport.

The major problem is that they have used B-lymphoma cells, which is a cancerous cell line. The cancerous cell lines possess tumor microtubes. The characterization of nanotubes and microtubes are not done in the study. The majority of the tubes are micrometre in diameter. Diameter of the tubes were not mentioned anywhere in the study. None of the videos shows the real transfer of the molecules through these tubes. Tumor microtubes are close-ended tubes where one can see mitochondria traveling in a bi-directional way. Tubes were not even characterized on the basis of hovering nature. Without the proper characterization of TNTs, the study is not of the quality to accept.

Comments on the Quality of English Language

NA

Author Response

Response to Reviewer 2 Comments

We appreciate the time and effort that you have dedicated to provide your valuable feedback on our manuscript. While we are grateful for the comments on our paper, we respectfully disagree with some points.

Here is a point-by-point response to the Reviewer’s comments and concerns.

Point 1: The major problem is that they have used B-lymphoma cells, which is a cancerous cell line. The cancerous cell lines possess tumor microtubes.

Response 1: Both tunnelling membrane nanotubes and tumour microtubes are filopodium-like structures [1]. However, tumour microtubes are significantly thicker than tunnelling membrane nanotubes (average 1.7 μm vs. less than 1.0 μm), longer (several hundreds of μm, up to 500 μm vs. 30 μm on average) and possess a much longer lifespan (even more than 200 days vs. a few hours) [2]. Moreover, tumour microtubes were identified only in the brain in glioblastoma, promoting the progression of glioma by facilitating tumour dissemination [1–3].

Although A20 cells, derived from a B cell lymphoma, are cancerous, they constitute a liquid tumour, bearing no similarity to solid tumours like glioblastomas. Furthermore, the A20 cell line is a widely accepted model in immunology research, and its ability to form tunnelling membrane nanotubes is well known [4–9].

Point 2: The characterization of nanotubes and microtubes are not done in the study. The majority of the tubes are micrometre in diameter. Diameter of the tubes were not mentioned anywhere in the study.

Response 2: Tunnelling membrane nanotubes of the A20 B cell line were thoroughly characterised in our previous studies [10–13], which we refer to in the appropriate places in our manuscript. We have already identified the major factors influencing or determining their growth [10,11,13] and some functions of A20 B cell tunnelling membrane nanotubes [10,12]. Furthermore, we have published the morphological features of these cell projections, including their diameter, length, and frequency of formation [10,13].

Point 3: None of the videos shows the real transfer of the molecules through these tubes. Tumor microtubes are close-ended tubes where one can see mitochondria traveling in a bi-directional way. Tubes were not even characterized on the basis of hovering nature. Without the proper characterization of TNTs, the study is not of the quality to accept.

Response 3: In our initial study on tunnelling membrane nanotubes [10], we found that the nanotubes are not only enriched in mitochondria (see reference [10] Figure 3), but they also move within nanotubes from one cell to the neighbouring one (see reference [10] Supplementary Movie S3). We have also demonstrated that the nanotubes of the A20 cell line are open-ended hovering intercellular structures (see reference [10] Fig 1C).

References

  1. Casas-Tintó; Portela Cytonemes, Their Formation, Regulation, and Roles in Signaling and Communication in Tumorigenesis. Int J Mol Sci 2019, 20, 5641, doi:10.3390/ijms20225641.
  2. Wang, X.; Liang, J.; Sun, H. The Network of Tumor Microtubes: An Improperly Reactivated Neural Cell Network With Stemness Feature for Resistance and Recurrence in Gliomas. Front Oncol 2022, 12, doi:10.3389/fonc.2022.921975.
  3. Osswald, M.; Jung, E.; Sahm, F.; Solecki, G.; Venkataramani, V.; Blaes, J.; Weil, S.; Horstmann, H.; Wiestler, B.; Syed, M.; et al. Brain Tumour Cells Interconnect to a Functional and Resistant Network. Nature 2015, 528, 93–98, doi:10.1038/nature16071.
  4. Černý, J.; Stříž, I. Adaptive Innate Immunity or Innate Adaptive Immunity? Clin Sci 2019, 133, 1549–1565, doi:10.1042/CS20180548.
  5. Shi, Y.; Lu, Y.; You, J. Antigen Transfer and Its Effect on Vaccine-Induced Immune Amplification and Tolerance. Theranostics 2022, 12, 5888–5913, doi:10.7150/thno.75904.
  6. Zhu, C.; Shi, Y.; You, J. Immune Cell Connection by Tunneling Nanotubes: The Impact of Intercellular Cross-Talk on the Immune Response and Its Therapeutic Applications. Mol Pharm 2021, 18, 772–786, doi:10.1021/acs.molpharmaceut.0c01248.
  7. Stögerer, T.; Silva-Barrios, S.; Carmona-Pérez, L.; Swaminathan, S.; Mai, L.T.; Leroux, L.-P.; Jaramillo, M.; Descoteaux, A.; Stäger, S. Leishmania Donovani Exploits Tunneling Nanotubes for Dissemination and Propagation of B Cell Activation. Microbiol Spectr 2023, 11, doi:10.1128/spectrum.05096-22.
  8. Drab, M.; Kralj-Iglič, V.; Resnik, N.; Kreft, M.E.; Veranič, P.; Iglič, A. Formation Principles of Tunneling Nanotubes. In; 2023; pp. 89–116.
  9. Cordero Cervantes, D.; Zurzolo, C. Peering into Tunneling Nanotubes—The Path Forward. EMBO J 2021, 40, doi:10.15252/embj.2020105789.
  10. Osteikoetxea-Molnár, A.; Szabó-Meleg, E.; Tóth, E.A.; Oszvald, Á.; Izsépi, E.; Kremlitzka, M.; Biri, B.; Nyitray, L.; Bozó, T.; Németh, P.; et al. The Growth Determinants and Transport Properties of Tunneling Nanotube Networks between B Lymphocytes. Cellular and Molecular Life Sciences 2016, 73, 4531–4545, doi:10.1007/s00018-016-2233-y.
  11. Tóth, E.A.; Oszvald, Á.; Péter, M.; Balogh, G.; Osteikoetxea-Molnár, A.; Bozó, T.; Szabó-Meleg, E.; Nyitrai, M.; Derényi, I.; Kellermayer, M.; et al. Nanotubes Connecting B Lymphocytes: High Impact of Differentiation-Dependent Lipid Composition on Their Growth and Mechanics. Biochimica et Biophysica Acta (BBA) - Molecular and Cell Biology of Lipids 2017, 1862, 991–1000, doi:10.1016/j.bbalip.2017.06.011.
  12. Halász, H.; Ghadaksaz, A.R.; Madarász, T.; Huber, K.; Harami, G.; Tóth, E.A.; Osteikoetxea-Molnár, A.; Kovács, M.; Balogi, Z.; Nyitrai, M.; et al. Live Cell Superresolution-Structured Illumination Microscopy Imaging Analysis of the Intercellular Transport of Microvesicles and Costimulatory Proteins via Nanotubes between Immune Cells. Methods Appl Fluoresc 2018, 6, 045005, doi:10.1088/2050-6120/aad57d.
  13. Madarász, T.; Brunner, B.; Halász, H.; Telek, E.; Matkó, J.; Nyitrai, M.; Szabó-Meleg, E. Molecular Relay Stations in Membrane Nanotubes: IRSp53 Involved in Actin-Based Force Generation. Int J Mol Sci 2023, 24, 13112, doi:10.3390/ijms241713112.

We look forward to hear from you in due time regarding our submission and to respond to any further questions and comments you may have.